# Phytogeographic Characteristics of Montane Coniferous Forests of the Central Balkan Peninsula (SE Europe)

**DOI:** 10.3390/plants11233194

**Published:** 2022-11-22

**Authors:** Tijana Ilić, Nevena Kuzmanović, Snežana Vukojičić, Dmitar Lakušić

**Affiliations:** Institute of Botany and Botanical Garden “Jevremovac”, Faculty of Biology, University of Belgrade, Takovska 43, 11000 Belgrade, Serbia

**Keywords:** floristic richness, endemics, life forms, area types, coniferous forests, Balkan peninsula

## Abstract

We investigated taxonomic and endemic richness, patterns of spatial distribution, cenotic and spatial diversification, and chorological and life form spectra of montane coniferous forests in the central part of the Balkan Peninsula. We collected information on 1435 taxa (1351 at the level of species and 84 subspecies) with 65,289 species-occurrence data, published in 1930 original plots with a total area of about 215 ha in the analysis. All statistical analyses (univariate and multivariate) were performed on binary matrices prepared for different levels of analysis. Our main results showed that the montane coniferous forests of the central Balkan Peninsula represent very species-rich vegetation. At the same time, the high proportion of endemics indicated that the montane coniferous forests of the central Balkan Peninsula differ significantly from Central European and boreal forests of a similar type. Furthermore, we found that there were regional differences in the species composition of the coniferous forests of the Balkan Peninsula, and that the primary centers of floristic richness are located in the area of the central and continental Dinarides. This latter finding suggested that the true centers of the richness of European coniferous forests are located south of the Limestone Alps—Western Dinarides—Carpathian Foothills line in Romania, which used to be considered the center of the richness of the coniferous forests in Europe.

## 1. Introduction

Concerning the centers of world plant diversity, the Balkan Peninsula, as part of the Mediterranean basin, is recognized as one of the few extratropical biodiversity hotspots [1,2,3]. The main hotspots for species richness are confined to mountainous areas for European coniferous forests, particularly in the Calcareous Alps, the north-western Dinarides, and the Western Carpathians [4]. The present floristic diversity of the Balkan Peninsula is the result of heterogeneity of environmental factors, geological-historical changes, and human influences [5]. In the Balkan Peninsula and, especially, in the central part, various floristic influences from Central Europe, the boreal and arctic regions of Eurasia, and the central and east Mediterranean meet in the form of long-lasting, multidirectional migration processes of florogenesis from the Tertiary to the present, with emphasis on the Quaternary glacials [6,7]. As one of the major glacial refugia, the Balkan Peninsula was crucial in the formation of European flora and fauna [8,9,10].

The heterogeneity of environmental factors, as one of the strongest drivers of floristic diversity, is best represented by the diversity of habitat types found in the studied area. Since different plants inhabit different phytocoenoses, depending on their ecological and coenotic affiliation, knowledge of the floristic composition of the different vegetation types is crucial to understand the overall biodiversity of any area. In this sense, the exceptional richness of plant species recorded in the Balkan Peninsula is a consequence of the extraordinary diversity of its vegetation [11,12], represented by the presence of eleven of the fourteen zonal, and all five azonal, types of natural vegetation in Europe [13]. 

Most of the conifers living on the Balkan Peninsula belong to the group of Balkan endemics, which includes the following: *Abies borisii-regis* Mattf., *Abies cephalonica* Loudon, *Picea omorika* (Pančić) Purk., *Pinus peuce* Griseb., *Pinus nigra* subsp. *dalmatica* (Vis.) Franco, or Balkan–Apennine endemic, *Pinus heldreichii* Christ. The Balkan Peninsula conifers also include the Balkan–Asia Minor–Crimean sub-endemics, *Pinus nigra* subsp. *pallasiana* (D. Don) Holmboe. These facts, as well as the fact that only a few species have a broader European (*Abies alba* Mill., *Pinus nigra* J. F. Arnold subsp. *nigra*) or Eurasian distribution (*Picea abies* (L.) H. Karst., *Pinus sylvestris* L.), indicate that the coniferous forests of the Balkan Peninsula represent an extremely interesting biogeographical phenomenon, the floristic diversity of which has not yet been extensively analyzed.

In Europe, 46% of forests are predominantly coniferous [14]. They are mostly distributed in the taiga biome in northern and northeastern Europe and in the high mountain ranges, which belong to the temperate deciduous broadleaved forest biome, but they also exist in the EU-Mediterranean and Supra-Mediterranean areas of southern Europe [5,13,15].

In the Balkan Peninsula, mono- and oligo-dominant coniferous forests are common in the Mediterranean and temperate zones, from the coast to altitudes above 2000 m. All forests are differentiated into two major formations, Mediterranean–Supra-Mediterranean and temperate–boreo-montane [11,12,13]. According to Mucina et al. [12], the first formation includes the following: (a) Relict Supra-Mediterranean Hellenic fir and black pine montane forests (*Abietion cephalonicae* Horvat et al., 1974); (b) Supra-Mediterranean cypress forests of Crete (*Aceri sempervirentis-Cupression sempervirentis* Barbero et Quézel ex Quézel et al., 1993) and (c) Thermo–Meso-Mediterranean pine forests of the central and eastern Mediterranean (*Pinetalia halepensis* Biondi, Blasi, Galdenzi, Pesaresi et Vagge in Biondi et al., 2014). The second formation consists of: (d) Holarctic coniferous forests on oligo-trophic and leached soils at high altitudes in the mountains (*Vaccinio-Piceetea* Br.-Bl. in Br.-Bl. et al., 1939) and (e) Relict pine forests on calcareous and ultramafic substrates (*Erico-Pinetea* Horvat 1959).

Only six coniferous species characterize the montane coniferous forests of the central Balkan Peninsula. Norway spruce (*Picea abies*) and Scots pine (*Pinus sylvestris*) are widespread species in the boreal, temperate, and boreo–montane zones of Euroasia, reaching their southernmost distribution in the Balkan Peninsula; fir (*Abies alba*) is distributed in the high mountains of central and southern Europe. Bosnian pine (*Pinus heldreichii*) is a sub-endemic of the Balkan and Apennines, and Macedonian pine (*Pinus peuce*) and Serbian spruce (*Picea omorika*) are Balkan endemics [6,16]. Apart from the Norway spruce (*Picea abies*) and Scots pine (*Pinus sylvestris*), other conifer species studied here do not occur in the boreal coniferous forests of Europe [13].

In the context of the importance of biodiversity protection, coniferous forests have been identified as high-priority habitats. Thus, the Habitats Directive of the European Union, as the main legislative instrument in the field of nature conservation [17] in the Balkan Peninsula, recognizes seven European natural habitat types, including one priority habitat type which is in danger of disappearance and whose natural range mainly falls within the territory of the European Union (9530 * (Sub-)Mediterranean pine forests with endemic black pines).

Although there are numerous data collected during the intensive development of floristic and phytocenological science over the 20th century, which constitute a valuable source of information for the description, quantification, and analysis of biodiversity at local and regional scales [18,19], recent studies that systematically address various aspects of Balkan floristic diversity [6,7,19,20,21,22,23,24,25,26,27,28,29,30,31,32,33,34,35,36,37,38,39,40,41,42,43,44,45,46] did not pay full attention to the floristic diversity and phytogeography of the coniferous forests of the Balkan Peninsula. 

Considering that the Balkan Peninsula has received less attention from phytogeographers than other parts of Europe and that, in the case of the Balkans, simple figures of exceptional species richness do not illustrate the fundamental importance of the region in terms of its conservation value [47], this work aimed to investigate: (1) the taxonomic and endemic richness and diversity, (2) the patterns of spatial distribution, (3) the cenotic and spatial diversification, (4) the chorological and life-form spectra of montane coniferous forests in the central part of the Balkan Peninsula, thus, helping to demonstrate the uniqueness and distinctiveness of the floristic diversity and phytogeographic characteristics of this part of Europe, and (5) to emphasize the special value of this type of habitat in the protection of nature and biodiversity.

## 2. Materials and Methods

### 2.1. Study Area, Forest Types, and Data Gathering Principles

The area under study covers the territory of the Balkan Peninsula, from Snježnik Mountain in Croatia in the north-west to the Pelister Mountain in North Macedonia in the south and the Central Stara planina Mountain in the east. (Figure 1). The considered geographical area of the Balkans follows the phytogeographical boundaries given by Reed et al. [48], while the classification of the mountain systems of the Balkans follows Stevanović et al. [37].

For the purpose of analyzing various aspects of phytogeographic characteristics of mountain coniferous forests in the central part of the Balkan Peninsula, we adapted the concept of “operational ecological units” [49], where the unit on which analyses are performed is a plant formation that has a well-defined floristic composition, unique physiognomy, similar habitat conditions and, consequently, similar ecological functions. The analyses were conducted at three hierarchical levels, representing the ecological and geographical diversities of coniferous forest formations in the investigated area.

At the first level, we distinguished two basic groups that unite: (A) boreo-montane and subalpine spruce and pine forests, which, according to Bohn et al. [13], belong to the vegetation formation D, Mesophytic and hygromesophytic coniferous forests, and (B) Sub-Mediterranean orotemperate dry relict pine forests on carbonate and ultramafic substrates, which, according to Bohn et al. [13], belong to the vegetation formation K, Xerophytic coniferous forest. In the text, we used the composite name “dark spruce forest type (*Vaccinio-Piceetea*)” for the first group and “light pine forest type (*Erico-Pinetea*)” for the second.

Within these two basic forest types, we singled out six units at the second level that correspond to forest formations dominated by the main coniferous species that make up the forests of the Balkan Peninsula. Thus, within the dark coniferous forests, four units were singled out: Norway spruce forests (*Vaccinio-Piceion*), Scots pine forests (*Pinion sylvestris*), Serbian spruce forests (*Piceion omorikae*), and Macedonian pine forests (*Pinion peucis*). Within the light coniferous forests, two units were singled out: Bosnian pine forests (*Pinion heldreichii*) and ultramafic black pine forests (*Orno-Ericion*).

Finally, the third level defines the operational units that make up the geographical variants of coniferous forests, the occurrence of which is registered in the basic mountain systems on the Balkan Peninsula, namely, Dinarides, Scardo-Pindic mountains, Rhodope mountains, and Balkan (Stara planina) mountains. According to Stevanović et al. [37], mountains are classified into four mountain systems and 16 mountain groups. Hierarchical relationships, names, and codes of the operational units are listed in Table 1.

The names of plant formation used in this manuscript have no formal syntaxonomic meaning, but are used as associative names, corresponding to the different forest types.

The total estimated extent of the area considered for this study is 1,209,553 hectares, while the estimated area occupied by operational ecological units at Levels I and II is shown in Table 1. The evaluation of these areas was recalculated, based on “Map of Natural Vegetation of Europe in scale 1:2,500,000” [13].

In accordance with the adapted concept of plant formation as “operational ecological units”, we used vegetation data from original phytocoenological tables in research articles, book chapters, master’s theses, and dissertations, published between 1938 and 2012, as the source for species occurrence data (Appendix A), with data from seven publications exported from the Balkan Vegetation Database [50,51]. In total, data from 1930 plots with a total area of approximately 215 ha (2,147,291 m^2^) were included in the analyses. Although the area where the data on species occurring in coniferous forests were collected was small (only 2.15 km^2^), particularly when compared to the extent of the entire region, it was, at the same time, very representative. Namely, the size of the plots included in the analysis varied between 60 and 20,000 square meters, with an average of 986 m^2^, which corresponds to the area representing the floristic composition of forest stands and is commonly used for phytocenological research on forests. In addition, the 1930 plots, from which the species data were taken, are very well distributed over the range of mountain coniferous forests of the Central Balkan Peninsula (Figure 1).

In each plot, we grouped multiple records of the same species occurring in more than one vegetation layer (e.g., tree species occurring in the herbaceous, shrub, and tree layers), removed non-vascular plants, and counted the number of species. Finally, a dataset of 65,289 species-occurrence data was formed, on which various analyses were performed.

Since the distribution of records by year for the entire area had a normal distribution, and the distribution of records by year was essentially different for different mountain systems (Appendix A), all records were included in the analysis, regardless of possible bias from long-term vegetation changes.

For non-georeferenced, or inaccurately georeferenced, plots, secondary accurate georeferencing was performed in Ozi Explorer and on Google Earth. Centers of floristic richness were shown in MGRS maps of 50 × 50 km, based on the UTM projection [52].

Taxon concepts and nomenclature largely followed the Checklist for Central Europe adopted for the EuroVegChecklist expert system in the JUICE program [12]. For a smaller number of species, primarily restricted to the area of the Balkan Peninsula, the nomenclature generally followed the Euro + Med PlantBase.

For defining areal groups, we used the classification based on the distribution area types proposed by Meusel et al. [53,54] and Meusel and Jaeger [55], which were modified for the territory of Serbia by Stevanović [56].

The basic life forms of the plants were determined according to Raunkiaer [57], supplemented by Mueller-Dombois and Ellenberg [58] and Stevanović [56].

### 2.2. Statistical Analyses

All statistical analyses (univariate and multivariate) were performed on binary (presence–absence) matrices prepared for different levels of analysis.

Similarity and distance indices (Jaccard), as well as diversity indices, were calculated using the package Past v. 2.17 [59]. We used two indicators of diversity: species richness (a term more common when discussing biogeographical issues), to describe the absolute species number in the designated area, and the LogS/LogA index of species density, to describe the number of species per unit.

Cluster analysis (paired group with Jaccard distances) was performed using the package Past v. 2.17 [59], while principal coordinate analysis, using Jaccard distances, was performed using Canoco 5 [60].

## 3. Results and Discussion

### 3.1. Species Richness and Diversity

We collected information on 1435 taxa (1351 at the level of species and 84 subspecies) in the coniferous forests of the Balkan Peninsula, with 65,289 species-occurrence data. This number of taxa represented 22% of the estimated total number of vascular plant taxa in the Balkan Peninsula [36,47,61] and 12% of the total European flora [62] Considering that in the Alps, as a major European mountain system, approximately 5500 species have been recorded [62], the number of 1435 taxa recorded only in the coniferous forests of the mountain systems of the central part of the Balkan Peninsula represented a considerable species richness. 

Our results contradicted the previously expressed opinion that coniferous forests were relatively poor and uniform, in terms of species number and diversity [11]. In contrast, our results were consistent with research showing that European forests show an increase in alpha diversity, from the species-poor north-west to the species-rich south-east of Europe [4]. Fennoscandian forests are dominated only by Scots pine and Norway spruce (a recent colonizer) and contain a small number of shrubs, herbs, and ferns, and significantly more moss and lichen species than vascular plants [63]. In the White Carpathians, species richness decreases in coniferous forests [64]. 

Within the investigated area, the “plant species maxima” (number of species on the plot of 100 m^2^ and 1000 m^2^, according to Wilson [65]) were found in the unmanaged (natural) relict forests of Serbian spruce on Tara Mountain (71 on 100 m^2^) and Zvijezda Mountain (125 on 1000 m^2^). These results were consistent with the established general pattern that the alpha diversity of coniferous forests in Europe is highest in the mountains of central and south-eastern Europe (Calcareous Alps and adjacent north-western Dinarides, the Carpathian foothills in Romania, and the Western Carpathians in Slovakia), where the richest plots contain between 45 and 72 species [4]. However, the fact that the most species-rich plots in the studied forests had between 70 and 125 species, and that they were located in the refugial areas of the continental and south-eastern Dinarides, suggested that the true centers of diversity of European coniferous forests are located south of the line Calcareous Alps–Western Dinarides–Carpathian foothills in Romania. Although the plant species maxima of the coniferous forests of the Central Balkans were far below the 313 species per 1000 m^2^ in the Colombian tropical rainforest, or the 233 taxa per 100 m^2^ of the Costa Rican tropical rainforest, the floristic richness of these forests was greater than in some supposedly species-rich areas, e.g., the southwestern Australian forests or the Mediterranean heathland [65].

All the top hotspots for the coniferous forest species richness were in the Central Balkans, dominated by limestone or other calcareous bedrock types, which was consistent with the results of a study on alpha diversity of vascular plants in European forests [4], and a study on ecological indicator values, which suggested that the vast majority of Central European vascular plant species prefer base-rich and calcareous soils, as also applies for the forest flora [66,67]. Our results showed that floristic richness on a non-carbonate substrate was almost half that of the richness on carbonate, with a higher number of species inhabiting ultramafic (species maximum 76 per 1000 m^2^), rather than silicate substrates (species maximum 33 per 1000 m^2^). All this was consistent with the results that explained why the flora was very rich in biogeographical regions dominated by calcareous bedrocks [66,67,68,69] 

Although not measured directly, based on the geographic location and general topography of the terrain where the plots with plant species maxima were recorded (mountainous areas with pronounced slopes, peaks, karst fields, and mountain plateaus, as well as deep canyons and gorges), we could conclude that the most species-rich montane coniferous forests of the Central Balkans often grew on shallow soil and in rugged terrain, which was also consistent with patterns whereby more rugged terrain tends to harbor more species-rich forests than flat or gently undulating landscapes [4]. Terrain roughness influences increasing species richness in several ways. The most important are habitat heterogeneity and spatial mass effects [70], locally buffered climate change and associated refugial effects during periods of macroclimatic variability [71] and poor accessibility. The low management intensity and high stand age of mountain forests favor vascular plant richness in the understory (68).

The analyses at the forest types level I (Level I) showed that species richness (1212 taxa) for all taxa and endemics were higher in the dark spruce forest types (*Vaccinio-Piceetea*), while the percentage of endemic taxa (19%) and species density (logS/logA index) of 0.512 was higher in the light pine forest types (*Erico-Pinetea*) (Table 2, Appendix A). These values of species density were lower compared to the values of species density of the total number of vascular plants for individual European countries with a similar geographic position as the Balkan Peninsula, such as Spain (0.723) or Italy (0.684), which are areas with the greatest species richness in Europe [6], but higher in comparison with the species density for the same forest types only in the territory of Serbia, which has dark spruce forests 0.301 and light pine forests 0.335 [19]. Comparing the total number of species recorded in the coniferous forests of the Balkan Peninsula (1212 taxa in dark spruce forests and 861 taxa in light pine forests) with the number of species for the dark spruce forests in Serbia (703) and light pine forests (683) [19], it could be seen that there were regional variations in the species composition of the coniferous forests of the Balkan Peninsula. Similarly, the total number of endemic species recorded in the coniferous forests of the study area (dark spruce forests, 188, vs. light pine forests, 162) was higher than the number of endemics recorded in the same habitat types in Serbia (dark spruce forests, 100, vs. light pine forests, 125) [40].

The light pine forests in the Balkan Peninsula, with a percentage of 19% endemic taxa, had among the higher values within the mountain ranges in the Mediterranean region, where the endemism rate ranged from 10.2% in the Pindus Mountains in Greece to 28.18% on the Baetic Mountains in Spain [72]. Furthermore, the light pine forest types on the Balkan Peninsula and some other vegetation types (such as those of the classes *Asplenietea trichomanes*, *Festuco-Brometea*, etc.) were the richest in Balkan endemic taxa at specific and subspecific ranks in central Serbia and Kosovo and Metochia regions [40]. The estimated number of endemics in other massive European mountain systems showed that 7% of the flora was endemic to the Alps, 5% to the Pyrenees, 6% to the montane flora of Crimea [73], and 12% to the Carpathians [74].

The high percentage of endemic taxa in montane coniferous forests could be considered a natural consequence of the role of mountains as centers of speciation [75]. Moreover, calcareous mountains, in areas such as the Alps or the Iberian Peninsula, were richer in endemic species than acidic areas [76,77], and the karst of the Dinaric Mountains was the most extensive example of a calcareous mountain in Europe. In addition, the relatively weak and localized glaciation during the Pleistocene climate fluctuations [78,79] provided suitable environmental conditions for the long-term survival of various species and lineages, contributing to the high species diversity and endemism.

The analyses at the forest types level II (Level II) showed that Norway spruce forests (*Vaccinio-Piceion*), with 905, and Scots pine forests (*Pinion sylvestris*), with 807, recorded taxa that significantly stood out with more taxa than other forest types. Norway spruce forests are widespread throughout the Balkan Peninsula in a variety of edaphic factors and different habitats [11], both as zonal and secondary character forests [80], such as the widely planted forests in Croatia during the 19th century [81] or the diverse relict communities associated with edaphic factors and water conditions on the Kopaonik Mountain in Serbia [27]. The widest distribution and, thus, the greatest heterogeneity of environmental conditions certainly had a decisive influence on species richness in Norway spruce forests. Scots pine forests showed similar distribution and heterogeneity of habitats, so this could explain the high richness of species recorded in them. In contrast, the Serbian spruce forests (*Piceion omorikae*, 246) and Macedonian pine forests (*Pinion peucis*, 282) had significantly fewer taxa than the other forest subtypes (Table 2, Appendix A). These two forest subtypes covered the smallest areas in the study area and had the most uniform environmental habitat conditions, which could be the reason why their species richness was significantly lower compared to Norway spruce and Scots pine forests. Moreover, Macedonian pine forests are located in the southernmost areas of zonal boreo-montane coniferous forests, and, specifically, in the drier parts that lie between temperate continental and Mediterranean mountain climates. Unlike Macedonian pine, Norway spruce is sensitive to summer drought, resulting in an unfavorable water regime that makes survival impossible for many species [11]. Serbian spruce is a poor competitor and inhabits extreme sites, such as cliffs and peat bogs [11,82], where only a few species can survive. Moreover, Serbian spruce forests are very localized, do not occupy large areas, and are severely degraded by anthropogenic factors and forest fires [83]. At the same time, Macedonian pine forests had one of the highest percentages of endemic taxa (18%) and the lowest logS/logA index (0.442), indicating the exceptional biogeographic importance of these forests.

In comparison, Serbian spruce forests had the lowest percentage of endemic taxa (9%) and the highest logS/logA index (0.544). This ratio was consistent with Peñas’s pattern for endemism in Mediterranean mountains, where low species richness is followed by a high percentage of endemics and vice versa [73]. However, Bosnian pine forests (*Pinion heldreichii*) were an exception to this observation. These forests had the highest percentage of endemic taxa (20%) and, at the same time, a high number of all taxa (579) and endemic taxa (114).

Bosnian pine is a heliophyte and xerophyte species [84], which makes these forests open and light, enabling the development of herbaceous and shrub layers. Bosnian pine forms azonal communities [11] and also forms a wide range of other different communities, such as mixed and transitional, with other community types, inhabiting rocky substrates, forests, mountains above the sea, and sites deeper inland [85,86,87,88,89]. A variety of environmental conditions, as well as a relict character of the species and the communities themselves, could also be the reason for the extraordinary species richness in Bosnian pine forests.

The ultramafic pine forests (*Orno-Ericion*) also had an increased number of all taxa (531) but had the lowest percentage of endemics (13%). These azonal pine forests occupy unique habitats with extreme soil–microclimate relationships, such as ultramafic soils. They are relict forests with an open canopy that allows the growth of the lower plant layers [11].

At the regional level (Level III), the highest species richness was recorded in the Norway spruce (Vacc Pic Din. 841) and Scots pine (Pin syl Din, 686) forests of the Dinaric mountain system, while the lowest number of species was recorded in the Scots pine forests (Pin syl Balk, 62) and Macedonian pine forests (Pin peuc Balk, 79) of the Balkan mountain range (Table 2, Appendix A). There was a significant disparity between the data collected in this research for the alpha diversity of Norway spruce forests in the different mountain systems. One possible reason for this could be the difference in the scope of the surveys conducted and the data availability. There are conflicting opinions in the relevant bibliographic data. Some refer to the uniformity of the Norway spruce forests of south-eastern Europe with those in Central Europe, the Alps, and Fennoscandia [11,82]. In contrast, others claim that the nemoral montane coniferous forests harbor a whole range of autochthonous temperate and sub-meridional mountain plants that do not occur in the boreal regions and the east European lowlands [13]. Therefore, we propose a further and more detailed study of the floristic diversity and differentiation of Norway spruce forests in south-eastern Europe.

The highest species density (LogS/LogA) was found in the Serbian spruce forests (Pic omor Din, 0.544) and Scots pine forests (Pin syl Din, 0.512) of the Dinarides, while the lowest was found in the Macedonian pine forests (Pin peuc Rod, 0.395) of the Rhodope mountain system (Table 2, Appendix A). Finally, the highest percentage of endemic taxa was found in the Macedonian pine forests of Scardo-Pindic (Pin peuc ScPind, 19%) and the Bosnian pine forests (Pin-heldr Din, 18%) of the Dinaric mountains, while the lowest percentage of endemic taxa was found in the Scots pine forests (Pin syl Balk, 3%) of the Balkan Mountains.

### 3.2. Centers of Floristic Richness

In our study, the primary center of the floristic richness of coniferous forests was located in the western-central part of the Balkan Peninsula, within the following 50 × 50 MGRS squares: CP3 (507), DN1 (419), and CP4 (402) (Figure 2, Appendix A), while the secondary centers of richness were located in the same part of the Balkan Peninsula, in the squares: DP2 (357), XK2 (347), DN3 (337), CP1 (336), DN2 (323). As expected, the lowest number of species was recorded in the MGRS squares on the south-eastern (GL1—17, LG2—20), eastern (LH2—46), and south-western (CN2—20, BN3—38, BN4—44) borders of the study area (Figure 2).

Concerning the mountain systems (Figure 3a), the incomparably greatest number of species was recorded in the Dinaric Mountains (1273) and the least in the Balkan Mountains (240) and the Scardo-Pindic Mountains (277). Regarding the individual mountain groups, the most species-rich groups were also within the Dinaric system. The mountains of the Tara group (853) had the highest number of taxa, followed by the mountain groups of Bjelašnica (453) and Kopaonik (412). The lowest number of taxa was recorded in the mountains of the Orjen group (44), western Stara Planina group (74), Suva Planina group (89), Pelister 102, and western Rhodopes 105 (Figure 3b).

Compared to previous findings on centers of various aspects of floristic diversity in the Balkan Peninsula, our results look quite different at first glance. Namely, a few comparative studies dealing with the distribution and diversity centers of the total flora [6], Balkan endemics [30,32,36,40,44], Orophytic [7], Arctic-Alpine [37], and Boreo-Montane plants [43], showed that the highest mountains of the central part of the Balkan Peninsula, with altitudes exceeding 2000 m, such as Prokletije, Durmitor, Šar planina, Korab, Koritnik, Paštrik, Kopaonik, Stara planina, Rila, Pirin, etc., represented the primary centers of floristic diversity. The same studies showrf that mountains of medium altitude (between 1000 and 2000 m a.s.l.), such as Tara, Zlatibor, Besna Kobila, Strešer, etc., were secondary centers of floristic diversity.

The finding that the primary centers of the floristic diversity of coniferous forests were in the mountains having medium altitude (between 1000 and 2000 m a.s.l.) in the area of the central and the continental Dinarides was consistent with the general ecology, distribution, and diversity of coniferous forests in the study area. In fact, in the area of the Tara mountain group, which includes Tara, Zvijezda, Zlatibor, Zlatar, Golija, Pobijenik, Jadovnik, Ozren, and the Pešterska visoravan plateau, there were the largest areas of well-preserved boreo-montane and relict oroclimatic forests, as well as the greatest diversity of environmental conditions, which was reflected in the presence of different variants of Norway spruce forests (*Vaccinio-Piceion*), Scots pine forests (*Pinion sylvestris*), Serbian spruce forests (*Piceion omorikae*), ultramafic black pine forests (*Orno-Ericion*), and small fragments of Bosnian pine forests (*Pinion heldreichii*). The forests in this area developed on dolomites, limestones, and ultramafics, on mountain slopes and plateaus, and in deep canyons. Moreover, and this is very important for coniferous forests, these forests developed in very favorable mountainous conditions with a temperate humid climate. In addition to these ecological reasons, undoubtedly a significantly higher number of species recorded in this part of the study area could be associated with the considerably better research status of coniferous forests in this part of the Balkan Peninsula.

### 3.3. Cenotic and Spatial Diversification

The principal coordinate analysis (PCoA) for the main forest subtypes (level II) showed that the analyzed groups were well differentiated (Figure 4a). The highest level of specificity was shown by the Serbian spruce forests (*Piceion omorikae*), which were located in the positive parts of the first and second axes, and the Macedonian pine forests (*Pinion peucis*), which were located in the positive part of the first axis and in the negative part of the second axis. A high level of specificity was also shown by the Ultramafic pine forests (*Orno-Ericion*), which were located in the positive part of the second axis and in the negative part of the first axis, while the remaining three subtypes were positioned in different parts of the quadrant bound by the negative parts of the first and second axes (Figure 4a).

The cluster analysis yielded the same relationships as the previous analysis. However, based on the similarity index, which exceeded 50% for all pairs, it could be concluded that all forest subtypes were well-defined floristically (Figure 4b).

Based on the obtained lists of exclusive taxa (taxa occurring in only one group and not present in any other analyzed group), it could also be concluded that all forest subtypes were well-defined floristically (Table 3). The highest number of exclusive taxa was recorded in the Norway spruce forests (*Vaccinio-Piceion*, 221), Scots pine forests (*Pinion sylvestris*, 133), and Bosnian pine forests (*Pinion heldreichii*, 118). In the ultramafic pine forests (*Orno-Ericion*), 97 exclusive taxa were recorded, while in the Macedonian pine forests (*Pinion peucis*) 28 taxa were recorded, and in the Serbian spruce forests (*Piceion omorikae*) only six taxa were found. The most important exclusive species for all subtypes of the analyzed forests are listed in Table 3.

Such differentiation of the studied forests was generally consistent with the formal phytocoenological classification of these phytocoenoses, which are defined as syntaxa at the level of alliances: *Piceion excelsae* Pawłowski et al., 1928 and *Abieti-Piceion* (Br.-Bl. in Br.-Bl. et al., 1939) Soó 1964, *Dicrano-Pinion sylvestris* (Libbert 1933) W. Matuszkiewicz 1962, *Pinion peucis* Horvat 1950, *Erico carneae-Piceion omorikae* Mucina et Čarni 2016, *Pinion heldreichii* Horvat 1946 and *Erico-Fraxinion orni* Horvat 1959 [12]. However, the relationships between the analyzed forests identified in our analyses did not correspond to the higher classification proposed in the most recent hierarchical floristic classification system of the Vegetation of Europe, proposed by Mucina et al. [12]. First of all, our operational units *Piceion omorikae*, *Pinion heldreichii*, and *Orno-Ericion*, classified according to Mucina et al. [12] in the common order *Erico-Pinetalia* Horvat 1959, class *Erico-Pinetea* Horvat 1959, occupied very distant places in PCoA space (Figure 4a), indicating very important differences in their floristic composition. Similarly, the very isolated position of our operational unit *Pinion peucis* in relation to the unit *Vaccinio-Piceion* with which it is classified in the common order *Piceetalia excelsae* Pawłowski et al., 1928 of *Vaccinio-Piceetea* class Bl. in Br.-Bl. et al., 1939 [12] indicated a very specific floristic composition of the forests of the Macedonian pine. Finally, the grouping of our operational units *Pinion heldreichii*, *Vaccinio-Piceion*, and *Pinion sylvestris* in a common quadrant in the PCoA diagram (Figure 4a) and a common cluster in the cluster diagram (Figure 4b), indicatedthat these three units were floristically most similar to each other, suggesting that their floristic relationships were not consistent with their formal syntaxonomic classification into two classes, *Erico-Pinetea* (*Pinion heldreichii*) and *Vaccinio-Piceetea* (*Vaccinio-Piceion*, *Pinion sylvestris*).

This considerable discrepancy between the established floristic relationships and the formal phytosociological classification at the highest hierarchical levels is almost certainly a consequence of the methodological approach to the syntaxonomic classification of vegetation. Namely, while our floristic analysis gave equal importance to all recorded species, phytosociological analyses attach particular importance to dominant, constant, and, especially, diagnostic species, so the differences noted are indeed due to different methodological approaches (floristic vs. phytosociological).

The principal coordinate analysis (PCoA) for forest subtypes at the regional level (Level III) revealed three main geographic groups (Figure 5a). The first group consisted of the Scots pine forests of the Balkan mountain system (Ps_B), which were clearly different from all other groups. The second group was located in the positive part of the first axis. It consisted of all forest types of the Dinarides, except the Macedonian pine forests (Pin_peuc_Din), which were located in the third group that included all forest types of the Scardo-Pindic, Rhodope, and Balkan mountain systems. Within the third group, a fine substructure could be observed, in which the forests of the Rhodope system stood out as a particular subunit (Figure 5a).

The cluster analysis revealed similar relationships to those identified in the PCoA analysis (Figure 5b). Namely, the cluster analysis identified three main groups of forest subtypes: (I) Scots pine forests of the Balkan mountains (Pin_syl_Balk); (II) Dinaric Scots pine forests (*Pinion sylvestris*—Pin_syl_Din), Norway spruce forests (*Vaccinio-Piceion*—Vacc_Pic_Din), Bosnian pine forests (*Pinion heldreichii*—Pin-heldr_Din) and ultramafic black pine forests (*Orno-Ericion*—Orn_Eric_Din), and (III) Other forest types of the Scardo-Pindic, Rhodope and Balkan mountain systems, which included the Dinaric forests of Serbian spruce (*Piceion omorikae*—Pic_omor_Din) and Macedonian pine (*Pinion peucis*—Pin_peuc_Din). Within the last group, the forests of the Rhodope system formed a separate subgroup (Figure 4b). The difference with regard to the results of the PCoA analysis was reflected only in the position of the Macedonian pine forests from the Dinaric mountains, which were found in the cluster analysis in the third mixed group, which included forests from the mountains of the southern and eastern Balkans.

As in the cluster analysis at level II, the similarity index at the regional level (level III) was above 50% for all pairs, confirming the previous conclusion that all forest subtypes were well-defined floristically (Figure 5b).

The obtained relationships between the analyzed forests at the regional level (Level III) showed an even greater discrepancy compared to the formal higher classification proposed in the latest hierarchical floristic classification system of the Vegetation of Europe, proposed by Mucina et al. [12]. Namely, these results showed that floristically homogeneous groups, with minor exceptions, formed distinct forest types that developed on the same mountain systems. This meant that the same forest types on different mountain systems differed from each other compared to other forest types located on the same mountain system. This result indicated that in the formation of the floristic composition of the studied coniferous forests, the geographical factor, namely, spatial distance, had a greater influence than any ecological differences that affected the formation of different phytocoenoses.

### 3.4. Chorological Spectrum

The chorological analysis of the total flora of the mountain coniferous forests of the central part of the Balkan Peninsula showed that, despite the differences observed, the dark spruce forest types (*Vaccinio-Piceetea*) and the light pine forest types (*Erico-Pinetea*) had almost identical horological structures (Figure 6a), characterized by a marked dominance of the Eurasian (EURAS), Eurasian mountain (EAM) and Central European (CE) chorological groups, which together accounted for over 75% of the total horological spectrum. Minor differences were found in the relationships among the main horological groups. For example, the dark spruce forest types (*Vaccinio-Piceetea*) were dominated by the Eurasian chorological group (EURAS—362 taxa; 30.04%), while the light pine forest types (*Erico-Pinetea*) were dominated by the Eurasian mountain chorological group (EAM—266 taxa; 31.11%). The Central European chorological group (CE) had a slightly higher proportion of the chorological spectrum in dark spruce forests (CE—213 taxa; 17.68%) compared to light pine forests (CE—128 taxa; 14.97%). On the contrary, a relatively low number of elements from the northern regions was registered in both basic forest types (Boreal—BOR—VP = 5.48% vs. EP = 4.80%, Arctic-Alpine—AA—VP = 1.99% vs. EP = 1.99%), as well as a relatively high number of elements from the southern regions in these mountain forests (Mediterranean–Sub-Mediterranean—MED-SUBM—VP = 8.22% vs. EP = 10.29%).

The analyses at level II showed that all forest subtypes shared the same basic chorological structure registered at level I (Figure 6b). To be specific, the overview of the chorological groups showed that the Eurasian (EURAS) and the Eurasian mountain (EAM) chorological groups dominated in all chorological spectra and that the Central European (CE) chorological group ranked third in terms of the number of species. However, minor variations were observed in some forest subtypes. In particular, the proportions of Eurasian (EURAS) and Eurasian mountain (EAM) chorological groups differed in different subtypes. Thus, the Eurasian (EURAS) had a slightly higher proportion than the Eurasian mountain (EAM) in most subtypes, except for the Macedonian pine forests (*Pinion peucis*) and the Bosnian pine forests (*Pinion heldreichii*), where the Eurasian mountain chorological group (EAM) dominated with over 33% in the spectrum. A significant deviation from the general spectrum was also registered in the Serbian spruce forests (*Piceion omorikae*), which had a slightly higher number of Central European (CE 23.7%) and boreal (BOR—8.2%) chorological groups, and in the Macedonian pine forests (*Pinion peucis*) with boreal (BOR—8.2%) chorological groups and the ultramafic pine forests (*Orno-Ericion*), where the number of Mediterranean–Sub-Mediterranean species was significantly higher (MED-DUBM—11.5%).

The dominance of Eurasian and Eurasian-mountain chorological groups was expected, due to the studied sites’ geographic locations and orographic characteristics. They showed that the core of the flora in the coniferous forests in the Balkan Peninsula consisted of Eurasian and Eurasian-mountain plants. On the other hand, the data for the Central European chorological group pointed to solid relationships with the Central European Plain, which could be explained by the connections between the Balkan Peninsula mountains with the Central European Plain during the glacial periods [90].

The relatively low number of boreal (BOR) and arctic-alpine (AA) elements was mainly explained by the fact that the studied area was located at the southern border of cold coniferous forests, which were ecologically suitable for plants of northern regions. Therefore, most plants adapted to cold mountain habitats in the studied area belonged to the Eurasian mountain chorological group (EAM), which included many endemic plants of the Balkan mountains. Zupančič, in his works on spruce communities from Slovenia to Bulgaria, concluded that the number of boreal species in spruce forests decreased from the north-west towards the south-east of the Balkan Peninsula, while Balkan species increased [91,92,93,94], which was consistent with our results. A comprehensive study of boreo-montane flora in selected parts of the Balkan Peninsula was conducted by Vukojičić et al. [43], who concluded that most boreal species occurred in mire vegetation, and oak and beech forests, and to a lesser extent in coniferous forests.

The southern location of the study area, which could explain the low number of boreal (BOR) and arctic-alpine (AA) elements, was, at the same time, the main reason for the high proportion of the Mediterranean–Sub-Mediterranean chorological group (MED-SUBM) in the total chorological spectrum. The significant Mediterranean floristic and florogenetic influences of the Mediterranean region on the formation of the flora of the Balkan Peninsula, including its high mountains, are well known and well documented [7]. Moreover, the specific disproportions in the participation of boreal and Mediterranean–Sub-Mediterranean species in coniferous forests might also be due to degradation processes under the strong anthropogenic influences of the recent past [27].

### 3.5. Endemism by Vegetation Types

The analysis of the chorological characteristics of the endemic flora of the analyzed forests (Figure 6c) showed that the two basic forest types (level I) had an almost identical chorological structure, dominated by endemic species with a broad Balkan distribution (eu-balk—VP = 5.20% vs. EP = 5.50%) and Illyrian endemics (illyr—VP = 3.14% vs. EP = 5.0%). In addition, the Moesian endemics (moes—VP = 1.32% vs. EP= 2.31%) and the Balkan–Carpathian sub-endemics (balk-carp—VP = 2.15% vs. EP = 2.7%) were also represented by a significant proportion. This complex structure of the chorology of endemics showed a transitional phytogeographic character corresponding to the geographic position of the studied mountains. It also indicated the floristic influence of the Carpathians, which could be explained by the processes of florogenesis in the Diluvium period [7]. As expected, Balkan–Anatolian (balk-anatol VP = 0.50% vs. EP = 0.50%) and Balkan–Pontic sub-endemics (balk-pont VP = 0.41% vs. EP = 0.50%) had a small contribution to these forests. This analysis also revealed nuanced differences in endemic plant structure between dark spruce (*Vaccinio-Piceetea*) and light pine (*Erico-Pinetea*) forest types, in that, the proportion of Illyrian and Moesian endemics was significantly higher in the light pine forest types. This indicated significant floristic influences from both provinces and defined the light pine forests as transitional vegetation types.

The analyses of endemism at level II (Figure 6d) showed that all forest subtypes were dominated by endemics with broad Balkan distribution (EU-balk—3.7% to 7.1%). Significant deviations from the general spectrum of endemics were shown only in Serbian spruce forests (*Piceion omorikae*), characterized by the complete absence of Moesian endemics and with the number of Balkan–Alpine sub-endemics being higher than that of Balkan–Carpathian sub-endemics (1.2% vs. 0.4%). Macedonian pine forests (*Pinion peucis*) were characterized by a higher proportion of Moesian compared to Illyrian endemics (2.8% vs. 1.1%), which was expected, due to the geographical location of Macedonian pine forest distribution.

### 3.6. Life Form Spectrum

The life form analysis revealed the absolute dominance of hemicryptophytes for both basic forest types (Figure 7a). The dark spruce forests (*Vaccinio-Piceetea*) had 56% hemicryptophytes and the light pine forests (*Erico-Pinetea*) had 65.1%. In contrast, scandentophytes (S) had an absolutely minimal value with less than 1% participation in the life form spectra of both forest types. Therophytes also had a low proportion in these forests (T—VP = 6.5% vs. EP= 5.0%), while the proportion of other life forms (P, Ch, G) varied by forest type. For example, geophytes (G) were the second most abundant life form in dark spruce forests with 13.8%. A similar proportion of geophytes was found in other forest types in Serbia [95]. In light pine forests, chamaephytes (Ch) accounted for 12.5%, which might indicate a high presence of xherotherm species [96]. There were also differences in tree life forms (P), which accounted for 11.1% in dark spruce forests and 8.5% in light pine forests (Figure 7a), which might be related to the degradation of light pine forests under strong anthropogenic negative influence [96].

The dominance of hemicryptophytes applied to all forest subtypes, from 42.9% in Serbian spruce forests (*Piceion omorikae*) to 58.5% in Macedonian pine forests (*Pinion peucis*) (Figure 7b). Serbian spruce forests differed from the other subtypes in the high proportion of phanerophytes, which were the next dominant with 21.6%, which was expected, since Serbian spruce mainly formed mixed, poly-dominant relict forests [95,97]. In the other forest subtypes, the life spectra were similar to those revealed in the analyses at level I.

In general, the life form spectrum of the mountain coniferous forests of the central Balkan Peninsula, with a predominance of hemicryptophytes and chamaephytes and low proportions of annuals and geophytes, was most similar to life form spectra of the Greek endemic plants [22], Balkan endemics in Bulgaria [32], and the Balkan endemic plants in central Serbia, and Kosovo and Metochia regions [40]. Here, it is important to emphasize that a large part of the analyzed endemic flora consisted of high-mountain plants, so we could speak of a specific high-mountain spectrum of life forms [40]. At the same time, the obtained life form spectra in mountain coniferous forests followed the general trends for the total flora of Serbia [95] and the Balkan Peninsula [98].

### 3.7. Conservation Value of Montane Coniferous Forests of the Central Balkan Peninsula

The main results of our research, showing that the mountain coniferous forests of the central Balkan Peninsula represent a unique biogeographical phenomenon in terms of floristic diversity, confirmed the well-known fact that coniferous forests generally have exceptional importance for biodiversity conservation. The importance of these forests for biodiversity conservation is primarily reflected in the fact that they represent one of the most important habitat types in ecological terms, hosting a large part of regional and global biodiversity [4,99].

In the Balkan Peninsula, the Habitats Directive of the European Union, as the main legislative instrument in the field of nature conservation [17], recognizes seven European natural coniferous habitat types: 9410 Acidophilous Picea forests of the montane to alpine levels (*Vaccinio-Piceetea*), 91BA Moesian silver fir forests, 9270 Hellenic beech forests with *Abies borisii-regis*, 9530 * (Sub-) Mediterranean pine forests with endemic black pines, 9540 Mediterranean pine forests with endemic Mesogean pines, 95A0 High oro-Mediterranean pine forests and 91R0 Dinaric dolomite Scots pine forests (*Genisto januensis-Pinetum*). These forest types require special measures to ensure the conservation of a wide range of rare, threatened or endemic animal and plant species. In the mountainous area of the Balkan Peninsula, which was the subject of our study, all types of coniferous forests of importance for the European Union were present, except Mediterranean pine forests with endemic Mesogean pines (9540) and Sub-Mediterranean pine forests with endemic black pines (9530).

Apart from the fact that these forests represent habitats that require special protection measures, the numerous species that build them also require special measures related to their populations. Among the species important for protection, the endemics, relicts and protected species stand out.

In our study, the presence of 219 endemic species and subspecies was registered in the mountain coniferous forests of the central Balkans, such as *Aconitum bosniacum*, *Alyssum markgrafii*, *Alyssum pirinicum*, *Amphoricarpos neumayeri*, *Aquilegia aurea*, *Aquilegia dinarica*, *Arabis croatica*, *Arabis ferdinandi-coburgii*, *Arabis scopoliana*, *Bornmuellera dieckii*, *Cardamine amara*, *Euphorbia gregersenii*, *Euphorbia montenegrina*, *Genista subcapitata*, *Haplophyllum boissieranum*, *Helleborus multifidus*, *Helleborus multifidus* ssp. *serbicus*, *Lathyrus binatus*, *Leucanthemum platylepis*, *Lonicera caerulea*, *Melampyrum heraleoticum*, *Peucedanum oligophyllum*, *Scrophularia bosniaca*, *Senecio thapsoides* ssp. *visianianus*, *Sesleria albicans* ssp. *angustifolia*, *Sesleria serbica*, *Spiraea cana*, *Stachys anisochila*, *Verbascum durmitoreum*, *Verbascum glabratum* ssp. *bosnense*, *Verbascum nicolai*, *Viola beckiana*, etc. (Appendix A). It is extremely important to emphasize that many of the conifers, which have a key role in the structure and functioning of the montane coniferous forests of the central Balkan Peninsula, are also Balkan endemics or sub-endemics (*Abies borisii-regis*, *Picea omorika*, *Pinus peuce*, *Pinus heldreichii*, *Pinus nigra* subsp. *pallasiana*), highlighting the unique conservation importance of these habitat types.

In addition, our research revealed the presence of 211 relict species in the same area, namely 67 glacial relicts (e.g., *Alchemilla glabra*, *Anthoxanthum alpinum*, *Arabis alpina*, *Arabis alpina* ssp. *alpina*, *Arctostaphylos alpinus*, *Aster alpinus*, *Carex atrata*, *Cerastium cerastoides*, *Cystopteris montana*, *Dryas octopetala*, *Epilobium anagallidifolium*, *Erigeron uniflorus*, *Gnaphalium norvegicum*, *Hieracium bifidum*, *Luzula spicata*, *Pedicularis verticillata*, *Persicaria vivipara*, *Phleum alpinum*, *Poa alpina*, *Salix reticulata*, *Saxifraga adscendens*, *Saxifraga aizoides*, *Saxifraga stellaris*, *Viola biflora*); 88 boreal relicts (e.g., *Adoxa moschatellina*, *Ajuga pyramidalis*, *Antennaria dioica*, *Arctostaphylos uva-ursi*, *Asplenium ruta-muraria*, *Asplenium viride*, *Avenella flexuosa*, *Betula pubescens*, *Blechnum spicant*, *Caltha palustris*, *Carex ornithopoda*, *Cicerbita alpina*, *Coeloglossum viride*, *Corallorrhiza trifida*, *Epilobium palustre*, *Epipogium aphyllum*, *Equisetum hyemale*, *Filipendula ulmaria*, *Galium boreale*, *Geum rivale*, *Goodyera repens*, *Listera cordata*, *Maianthemum bifolium*, *Moneses uniflora*, *Orthilia secunda*, *Parnassia palustris*, *Persicaria bistorta*, *Pyrola chlorantha*, *Pyrola media*, *Pyrola minor*, *Pyrola rotundifolia*, *Pyrola uniflora*, *Ribes alpinum*, *Rubus saxatilis*, *Trollius europaeus*, *Vaccinium myrtillus*, *Vaccinium uliginosum*, *Vaccinium vitis-idaea*) and 56 tertiary relicts (e.g., *Acer intermedium*, *Amelanchier ovalis*, *Aposeris foetida*, *Aremonia agrimonoides*, *Arnica montana*, *Aruncus dioicus*, *Asarum europaeum*, *Buphthalmum salicifolium*, *Calluna vulgaris*, *Convallaria majalis*, *Corylus colurna*, *Daphne alpina*, *Daphne blagayana*, *Daphne cneorum*, *Daphne laureola*, *Daphne mezereum*, *Daphne oleoides*, *Epimedium alpinum*, *Hacquetia epipactis*, *Hedera helix*, *Iberis sempervirens*, *Ilex aquifolium*, *Juglans regia*, *Lathraea squamaria*, *Ligustrum vulgare*, *Monotropa hypopitys*, *Omphalodes verna*, *Paris quadrifolia*, *Rhododendron ferrugineum*, *Ruscus hypoglossum*, *Sanicula europaea*, *Scopolia carniolica*, *Streptopus amplexifolius*, *Taxus baccata*, *Telekia speciosa*). All this confirmed the fact that the mountain coniferous forests of the central part of the Balkan Peninsula represent significant centers of endemism and, at the same time, are a strong refuge for many plants of the northern regions, which shifted their ranges to the Balkan Peninsula during the Pleistocene glaciations, as pointed out in [37,39,40,43,100,101]. The importance of the central Balkans is also underlined by the fact that postglacial migration from its refugia was one of the key factors shaping today’s high species richness of different forest types in the Alps and Central Europe [68,102]. At this point it should be added that numerous data confirm the existence of several refugia in the Balkan Peninsula (“refugia-within-refugia” model—Gómez and Lunt 2007 [103]). Within these smaller, isolated and ecologically distinct refugia, populations not only survived and maintained their genetic diversity, but also genetically differentiated [61,104]

Here it is important to emphasize that the glacial and boreal relicts on the Balkan Peninsula are located at the southern limits of their distribution, which further underlines their importance and necessity for biodiversity protection.

Finally, the great conservation importance of these conifer forests is also reflected in the fact that they are habitats for numerous internationally important species that are protected by key international legal documents related to nature protection. Thus, in our study, the presence of 10 taxa from the Habitats Directive of the European Union (*Arabis scopoliana*, *Arnica montana*, *Asplenium adulterinum*, *Echium russicum*, *Galanthus nivalis*, *Gentiana lutea*, *Gentiana lutea* ssp. *symphyandra*, *Pulsatilla grandis*, *Ramonda serbica*, *Tozzia carpathica*), 7 taxa from Resolution No. 6 of the Bern Convention [105](*Arabis scopoliana*, *Asplenium adulterinum*, *Echium russicum*, *Lilium jankae*, *Narcissus angustifolius*, *Pulsatilla grandis*, *Tozzia carpathica*) and 29 species listed in Appendix II of the CITES Convention [106] (*Cyclamen purpurascens* and 28 orchids, including very rare species, such as *Coeloglossum viride*, *Corallorrhiza trifida*, *Epipogium aphyllum*, *Goodyera repens*, *Gymnadenia odoratissima*, *Listera cordata*, *Listera ovata*) were recorded.

Apart from their undeniable importance for biodiversity protection, mountain coniferous forests of the Central Balkans, together with other forests at regional and global scales, also play an essential role in mitigating, and adapting to, the effects of climate change, as well as an essential function in the process of natural carbon sinks [99]. Unfortunately, past forestry management practices in the Balkan Peninsula did not meet the need to ensure high biodiversity in forest areas. Moreover, as part of the global forests, the Balkan forests are now facing new increased pressure related to the effects of climate change, which has led to extreme weather conditions (droughts, storms, fires) or more pests [99]. Therefore, unsustainable forestry practices should be prevented or corrected, and should be based on the EU Biodiversity Strategy for 2030 [107], and the New EU Forest Strategy for 2030 [108].

## 4. Conclusions

A number of 1435 taxa was recorded in the montane coniferous forests of the central Balkan Peninsula, which represents a considerable species richness.

“Plant species maxima” were found in the unmanaged (natural) relict forests of Serbian spruce on Tara Mountain (71 on 100 m^2^) and Zvijezda Mountain (125 on 1000 m^2^) in the continental Dinarides.

Our findings are consistent with previously known patterns of an increase in alpha diversity from the “species-poor” north-west to the “species-rich” south-east of Europe.

All of the top hotspots for the coniferous forest species richness are positioned in the central Balkan Peninsula and are dominated by limestone or other calcareous bedrock types, which is consistent with the results of the study of alpha diversity of vascular plants in European forests.

The most species-rich montane coniferous forests of the central Balkan peninsula often grow on shallow soil and rugged terrain, which is also consistent with the pattern that more rugged terrain tends to harbor more species-rich forests than flat or gently undulating landscapes.

The primary centers of the floristic diversity of coniferous forests in the central Balkan Peninsula are located in the area of the central and continental Dinarides, suggesting that the true centers of diversity of European coniferous forests are located south of the line Calcareous Alps–Western Dinarides–Carpathian foothills in Romania, which used to be considered the center of diversity of coniferous forests in Europe.

Species richness for all taxa and endemics groups is higher in the dark spruce forest types (*Vaccinio-Piceetea*), while the percentage of endemic taxa and species density (logS/logA index) is higher in the light pine forest types (*Erico-Pinetea*).

The light pine forests in the Balkan Peninsula, with a percentage of 19% endemic taxa, exhibited among the higher values within the mountain ranges in the Mediterranean region.

Concerning the mountain systems, the incomparably greatest number of species was recorded in the Dinaric Mountains (1273) and the least in the Balkan Mountains (240) and the Scardo–Pindic Mountains (277).

The analyzed groups of forests were well differentiated. The highest degree of specificity is shown by the Serbian spruce forests (*Piceion omorikae*) and the Macedonian pine forests (*Pinion peucis*), and to a lesser extent in the ultramafic pine forests (*Orno-Ericion*). In contrast, Norway spruce forests (*Vaccinio-Piceion*) and the Scots pine forests (*Pinion sylvestris*) show the greatest similarity.

Despite the observed differences, the dark spruce forest types (*Vaccinio-Piceetea*) and the light pine forest types (*Erico-Pinetea*) show an almost identical chorological structure, characterized by a marked dominance of the Eurasian (EURAS), Eurasian mountain (EAM), and Central European (CE) chorological groups, which together account for over 75% of the total chorological spectrum.

At the same time, a relatively low number of elements from northern regions (Boreal—BOR and Arctic–Alpine—AA) and a relatively high number of elements from southern regions (Mediterranean–Sub-Mediterranean—MED-SUBM) was recorded in the two basic forest types in investigated mountain coniferous forests.

The dark spruce forest types (*Vaccinio-Piceetea*) and the light pine forest types (*Erico-Pinetea*) also have an almost identical life form spectrum, characterized by a predominance of hemicryptophytes and chamaephytes and a small proportion of annuals and geophytes.

The mountain coniferous forests of the central Balkan Peninsula have exceptional importance in biodiversity conservation, since they represent the habitats of many important species. such as endemics, relicts, and nationally and internationally protected species.

Unfortunately, previous forest management practices in the Balkan Peninsula were not in line with the need to ensure high biodiversity in forest areas, so a significant number of species are endangered today. Therefore, unsustainable forestry practices should be prevented or corrected, and harmonized with the EU biodiversity strategy for 2030 and the New EU Forest Strategy for 2030.

## Figures and Tables

**Figure 1 plants-11-03194-f001:**
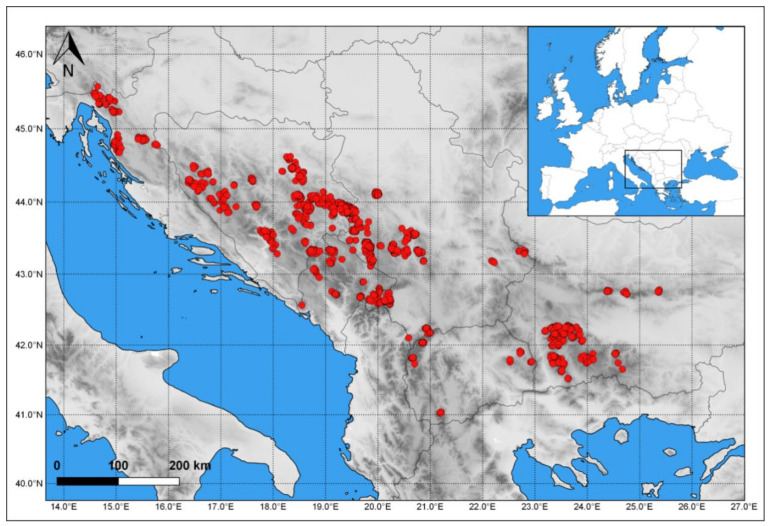
Distribution of coniferous forests in the Balkan Peninsula—position of the analyzed plots of coniferous forests.

**Figure 2 plants-11-03194-f002:**
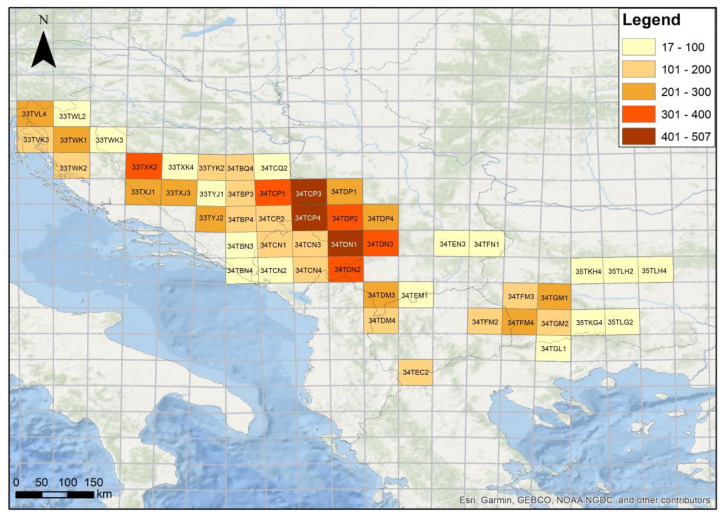
Centers of the floristic richness of coniferous forests in the Balkan Peninsula—number of taxa recorded within 50 × 50 MGRS squares.

**Figure 3 plants-11-03194-f003:**
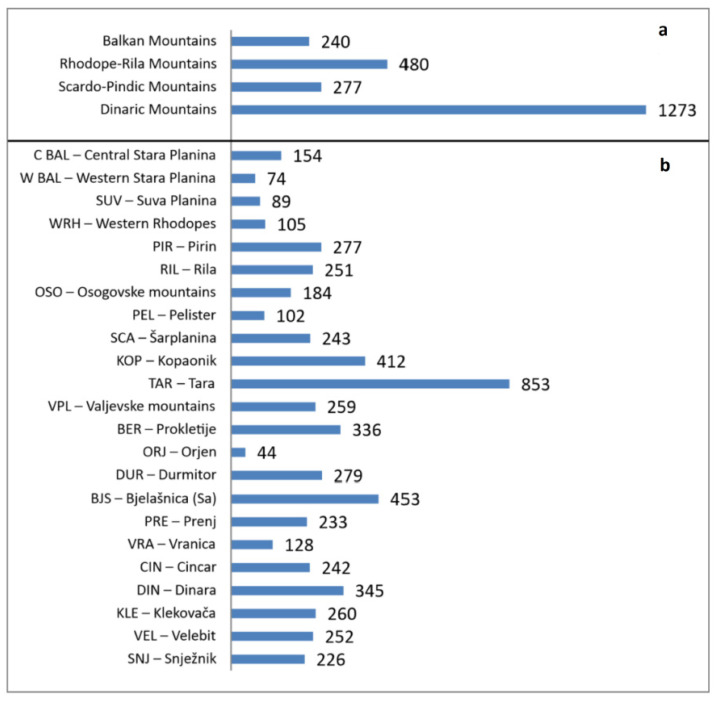
Number of species by mountain systems (**a**) and mountain groups (**b**).

**Figure 4 plants-11-03194-f004:**
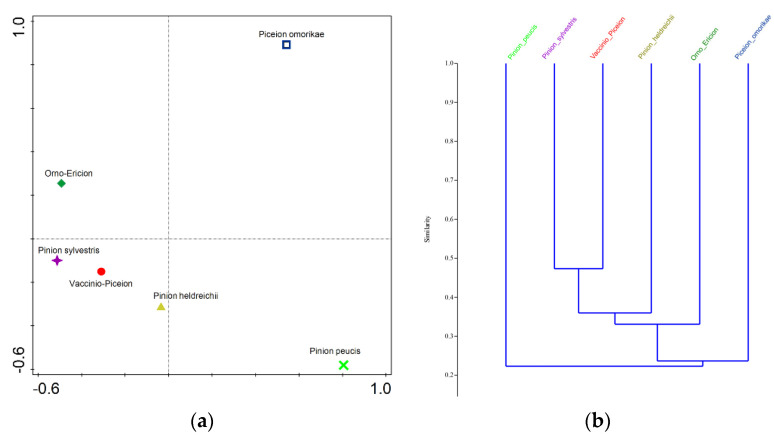
(**a**) Principal coordinate analysis (PCoA) for forest subtypes (Level II) (**b**) Cluster analysis for forest subtypes (Level II).

**Figure 5 plants-11-03194-f005:**
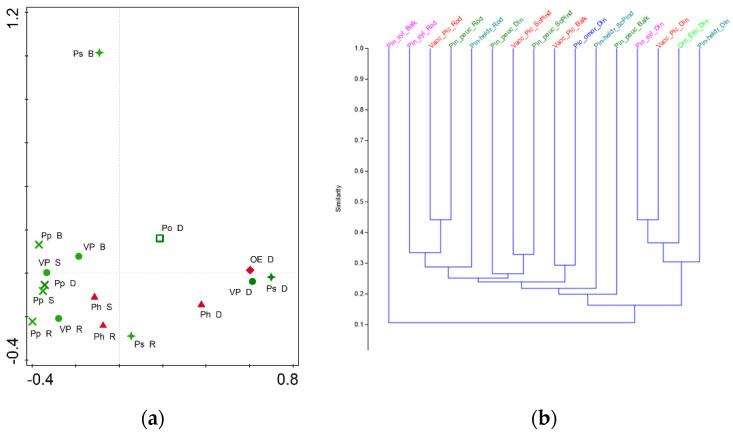
(**a**) Principal coordinates analysis (PCoA) for forest sub-types at the regional level (Level III) (**b**) Cluster analysis for forest sub-types at the regional level (Level III). VP D—*Vaccinio-Piceion* in Dinarides; VP S—*Vaccinio-Piceion* in Scardo-Pindic mountains; VP R—Vaccinio-Piceion in Rhodope mountains; VP B—*Vaccinio-Piceion* in Balkan mountains; Po D—*Piceion omorikae* in Dinarides; Ps D—*Pinion sylvestris* in Dinarides; Ps R—*Pinion sylvestris* in Rhodope mountains; Ps B—*Pinion sylvestris* in Balkan mountains; Pp D—*Pinion peucis* in Dinarides; Pp S—*Pinion peucis* in Scardo-Pindic mountains; Pp R—*Pinion peucis* in Rhodope mountains; Pp B—*Pinion peucis* in Balkan mountains; OE D—*Orno-Ericion* in Dinarides; Ph D—*Pinion heldreichii* in Dinarides; Ph S—*Pinion heldreichii* in Scardo-Pindic mountains; Ph R—*Pinion heldreichii* in Rhodope mountains.

**Figure 6 plants-11-03194-f006:**
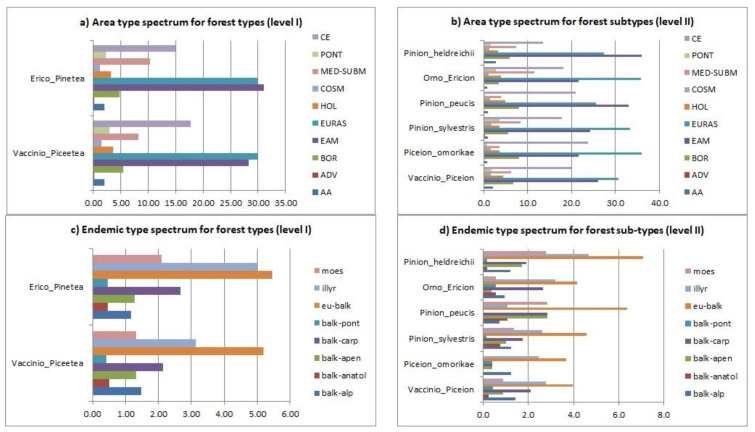
Chorological spectrum: (**a**) total flora of forest types (Level I), (**b**) total flora of forest subtypes (Level II); (**c**) endemics of forest types (Level I), (**d**) endemics forest sub-types (Level II).

**Figure 7 plants-11-03194-f007:**
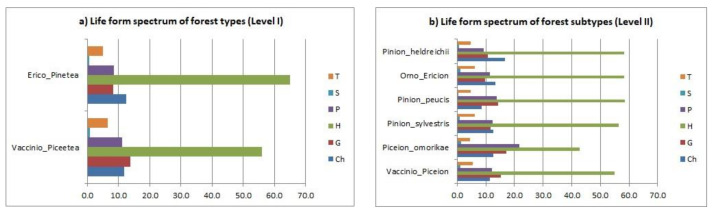
Life form spectrum of (**a**) forest types (Level I), (**b**) forest subtypes (Level II). Hemicryptophytes (H), Therophytes (T), Geophytes (G), Phanerophytes (P), Scandentophytes (S), Chamaephytes (Ch).

**Table 1 plants-11-03194-t001:** “Operational ecological units” and hierarchical levels of coniferous forests in the Balkan Peninsula for which the analyses were performed.

Level	Forest Types	Codes	Estimated Area of Occupancy (in ha)
**I**	**Dark spruce forest types (*Vaccinio-Piceetea*)**	** *Vaccinio-Piceetea* **	**1,059,824**
**II**	**Norway spruce forests (*Vaccinio-Piceion*)**	** *Vaccinio-Piceion* **	635,389
III	Norway spruce forests (*Vaccinio-Piceion*) in Dinarides	Vacc_Pic_Din	
III	Norway spruce forests (*Vaccinio-Piceion*) in Scardo-Pindic mountains	Vacc_Pic_ScPind	
III	Norway spruce forests (*Vaccinio-Piceion*) in Rhodope mountains	Vacc_Pic_Rhod	
III	Norway spruce forests (*Vaccinio-Piceion*) in Balkan mountains	Vacc_Pic_Balk	
**II**	**Scots pine forests (*Pinion sylvestris*)**	** *Pinion silvestris* **	318,880
III	Scots pine forests (Pinion sylvestris) in Dinarides	Pin_syl_Din	
III	Scots pine forests (Pinion sylvestris) in Rhodope mountains	Pin_syl_Rhod	
III	Scots pine forests (Pinion sylvestris) in Balkan mountains	Pin_syl_Balk	
**II**	**Serbian spruce forest (Piceion omorikae)**	** *Piceion omorikae* **	6697
III	Serbian spruce forest (Piceion omorikae) in Dinarides	Pic_omor_Din	
**II**	**Macedonian pine forests (Pinion peucis)**	** *Pinion peucis* **	98,858
III	Macedonian pine forests (Pinion peucis) in Dinarides	Pin_peuc_Din	
III	Macedonian pine forests (Pinion peucis) in Scardo-Pindic mountains	Pin_peuc_ScPind	
III	Macedonian pine forests (Pinion peucis) in Rhodope mountains	Pin_peuc_Rhod	
III	Macedonian pine forests (Pinion peucis) in Balkan mountains	Pin_peuc_Balk	
**I**	**Light pine forest types (*Erico-Pinetea*)**	** *Erico-Pinetea* **	149,730
**II**	**Bosnian pine forests (*Pinion heldreichii*)**	** *Pinion heldreichii* **	133,334
III	Bosnian pine forests (*Pinion heldreichii*) in Dinarides	Pin_heldr_Din	
III	Bosnian pine forests (*Pinion heldreichii*) in Scardo-Pindic mountains	Pin_heldr_ScPind	
III	Bosnian pine forests (*Pinion heldreichii*) in Rhodope mountains	Pin_heldr_Rhod	
**II**	**Ultramafic black pine forests (*Orno-Ericion*)**	** *Orno-Ericion* **	16,395
III	Ultramafic black pine forests (*Orno-Ericion*) in Dinarides	Orn_Eric_Din	

**Table 2 plants-11-03194-t002:** Relevé and formation data and diversity parameters for all three levels of research. Abbreviations: **S**—number of taxa; **logS/logA**—species density index; **S End**—number of endemic taxa. **Vacc Pic Din**—*Vaccinio-Piceion* in Dinarides; **Vacc Pic ScPind**—*Vaccinio-Piceion* in Scardo-Pindic mountains; **Vacc Pic Rhod**—*Vaccinio-Piceion* in Rhodope mountains; **Vacc Pic Balk**—*Vaccinio-Piceion* in Balkan mountains; **Pic omor Din**—*Piceion omorikae* in Dinarides; **Pin syl Din**—*Pinion sylvestris* in Dinarides; **Pin syl Rhod**—*Pinion sylvestris* in Rhodope mountains; **Pin syl Balk**—*Pinion sylvestris* in Balkan mountains; **Pin peuc Din**—*Pinion peucis* in Dinarides; **Pin peuc ScPind**—*Pinion peucis* in Scardo-Pindic mountains; **Pin peuc Rhod**—*Pinion peucis* in Rhodope mountains; **Pin peuc Balk**—*Pinion peucis* in Balkan mountains; **Orn Eric Din**—*Orno-Ericion* in Dinarides; **Pin heldr Din**—*Pinion heldreichii* in Dinarides; **Pin heldr ScPind**—*Pinion heldreichii* in Scardo-Pindic mountains; **Pin heldr Rhod**—*Pinion heldreichii* in Rhodope mountains.

Level I	No. Relevés	Relevé Area (m^2^)	S	Logs/Loga	S End	% End
*Vaccinio-Piceeetea*	1499	1,601,977	1212	0.497	188	16
*Erico-Pinetea*	431	545,314	861	0.512	162	19
**Level II**	**No. Relevés**	**Relevé area (m^2^)**	**S**	**logS/logA**	**S End**	**% End**
*Vaccinio-Piceion*	944	768,955	905	0.502	115	13%
*Piceion omorikae*	59	24,854	246	0.544	21	9%
*Pinion sylvestris*	348	463,215	807	0.513	108	13%
*Pinion peucis*	148	344,954	282	0.442	50	18%
*Orno-Ericion*	219	239,633	531	0.507	69	13%
*Pinion heldreichii*	212	305,681	579	0.504	114	20%
**Level III**	**No. Relevés**	**Relevé area (m^2^)**	**S**	**logS/logA**	**S End**	**% End**
Vacc Pic Din	758	617,444	841	0.505	89	11
Vacc Pic ScPind	26	21,179	121	0.481	14	12
Vacc Pic Rhod	124	101,007	194	0.457	19	10
Vacc Pic Balk	36	29,325	177	0.503	13	7
Pic omor Din	59	24,854	246	0.544	21	9
Pin syl Din	261	347,411	686	0.512	76	11
Pin syl Rhod	82	109,148	320	0.497	42	13
Pin syl Balk	5	6655	62	0.469	2	3
Pin peuc Din	21	48,946	114	0.439	13	11
Pin peuc ScPind	11	25,638	146	0.491	28	19
Pin peuc Rhod	95	221,423	129	0.395	15	12
Pin peuc Balk	21	48,946	79	0.405	11	14
Orn Eric Din	219	239,633	531	0.507	69	13
Pin heldr Din	114	164,376	433	0.505	77	18
Pin heldr ScPind	15	21,628	140	0.495	20	14
Pin heldr Rhod	83	119,677	236	0.467	39	17

**Table 3 plants-11-03194-t003:** Most important exclusive taxa for all subtypes of the analyzed forests in the Balkan Peninsula.

Spruce forests (*Vaccinio-Piceion*)	*Aconitum lycoctonum* ssp. *vulparia*, *Adenostyles glabra*, *Aquilegia nigricans*, *Asperula taurina*, *Caltha palustris*, *Cardamine trifolia*, *Cardamine waldsteinii*, *Chaerophyllum aromaticum*, *Chaerophyllum hirsutum*, *Circaea lutetiana*, *Cirsium waldsteinii*, *Clematis alpina*, *Coeloglossum viride*, *Dryopteris dilatata*, *Dryopteris expansa*, *Equisetum sylvaticum*, *Festuca altissima*, *Homogyne sylvestris*, *Hypericum umbellatum*, *Lamium orvala*, *Lathraea squamaria*, *Listera cordata*, *Lysimachia nemorum*, *Melampyrum barbatum*, *Myosotis scorpioides*, *Omphalodes verna*, *Petasites albus*, *Petasites hybridus*, *Phyteuma spicatum* ssp. *coeruleum*, *Poa hybrida*, *Ranunculus aconitifolius*, *Ranunculus acris*, *Ranunculus carinthiacus*, *Ranunculus platanifolius*, *Ranunculus serbicus*, *Rumex alpinus*, *Scopolia carniolica*, *Senecio papposus* ssp. *papposus*, *Silene alba*, *Soldanella dimoniei*, *Stachys dinarica*, *Symphytum tuberosum* agg., *Thelypteris limbosperma*, *Tilia platyphyllos*, *Trollius europaeus*, *Vicia oroboides* etc.
Serbian spruce forests (*Piceion omorikae*)	*Epipogium aphyllum*, *Equisetum hyemale*, *Euphorbia glareosa*, *Hierochloe australis*, *Petasites kablikianus* and *Saxifraga tridactylites*
Scots pine forests (*Pinion sylvestris*)	*Avenula planiculmis*, *Bupleurum apiculatum*, *Bupleurum praealtum*, *Campanula phrygia*, *Centaurea bracteata*, *Cerastium fontanum* ssp. *vulgare*, *Chrysopogon gryllus*, *Cirsium grecescui*, *Coronilla coronata*, *Coronilla vaginalis*, *Chamaecytisus polytrichus*, *Cytisus jankae*, *Euphorbia serpentini*, *Festuca bosniaca* ssp. *pirinensis*, *Festuca spadicea*, *Genista carinalis*, *Inula hirta*, *Laburnum anagyroides*, *Lathyrus alpestris*, *Linum serbicum*, *Narcissus radiiflorus*, *Orchis morio*, *Pedicularis comosa*, *Poa stiriaca*, *Potentilla regisborisii*, *Potentilla rupestris*, *Primula elatior*, *Quercus robur*, *Ranunculus pseudomontanus*, *Sanguisorba officinalis*, *Saxifraga blavii*, *Scleranthus perennis* ssp. *dichotomus*, *Stachys betonica*, *Stellaria palustris*, *Thlaspi ochroleucum*, *Veronica vindobonensis*, *Vicia cassubica* etc.
Macedonian pine forests (*Pinion peucis*)	*Aconitum burnatii* ssp*. pentheri*, *Campanula lanata*, *Crataegus orientalis*, *Crocus atticus*, *Festuca balcanica*, *Gentiana punctata*, *Geum reptans*, *Helleborus cyclophyllus*, *Jasione orbiculata*, *Knautia magnifica*, *Luzula spicata*, *Pedicularis friderici-augusti*, *Rhododendron ferrugineum*, *Senecio abrotanifolius* ssp. *carpathicus*, *Soldanella carpatica*, *Stachys tymphaea*, *Tozzia alpina*, *Viola orphanidis* etc.
Ultramafic pine forests (*Orno-Ericion*)	*Acinos hungaricus*, *Aristolochia pallida*, *Artemisia alba*, *Asplenium adulterinum*, *Astragalus onobrychis*, *Bupleurum ranunculoides*, *Calamagrostis villosa*, *Calamintha menthifolia*, *Capsella bursa-pastoris*, *Cardamine amara*, *Cardamine plumieri*, *Centaurea epapposa*, *Centaurea reichenbachii*, *Centaurea scabiosa* ssp. *spinulosa*, *Chamaecytisus austriacus*, *Chamaecytisus ciliatus*, *Chamaecytisus supinus*, *Cota tinctoria*, *Cytisus haeufelii*, *Dianthus sylvestris*, *Doronicum grandiflorum*, *Euphorbia glabriflora*, *Euphorbia gregersenii*, *Euphorbia montenegrina*, *Euphorbia virgata*, *Festuca rupicola*, *Fumana procumbens*, *Galium flavescens*, *Haplophyllum boissieranum*, *Isatis praecox*, *Lathyrus bauhinii*, *Linaria rubioides*, *Medicago falcata*, *Medicago prostrata*, *Ornithogalum umbellatum*, *Pedicularis brachyodonta*, *Peucedanum carvifolia*, *Phleum hirsutum*, *Phleum montanum*, *Phleum phleoides*, *Platanthera chlorantha*, *Poa chaixii*, *Podospermum laciniatum*, *Scleranthus perennis*, *Silene paradoxa*, *Stachys recta* ssp. *baldaccii*, *Thesium divaricatum*, *Thymus glabrescens*, *Tragopogon balcanicus*, *Viola beckiana* etc.
Bosnian pine forests (*Pinion heldreichii*)	*Alchemilla glaucescens*, *Amphoricarpos neumayeri*, *Anthyllis vulneraria* ssp. *alpestris*, *Arabis sudetica*, *Aster alpinus*, *Bornmuellera dieckii*, *Campanula hercegovina*, *Carlina biebersteinii*, *Centaurea achtarovii*, *Crepis dinarica*, *Draba lasiocarpa*, *Edraianthus tenuifolius*, *Festuca penzesii*, *Genista subcapitata*, *Gentiana clusii*, *Helleborus purpurascens*, *Iberis sempervirens*, *Leucanthemum chloroticum*, *Moltkia petraea*, *Poa macedonica*, *Poa media*, *Potentilla clusiana*, *Potentilla crantzii*, *Rhinanthus wagneri*, *Scabiosa graminifolia*, *Seseli kochii*, *Silene fabarioides*, *Thesium auriculatum* etc.

## Data Availability

The data presented in this study are available on request from the corresponding author. The data are not publicly available due to the database being part of a Ph.D. thesis. The database will be available after the publication of the dissertation.

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
