# Peer review of "Phytogeographic Characteristics of Montane Coniferous Forests of the Central Balkan Peninsula (SE Europe)"

_plants, 2022, doi:10.3390/plants11233194_

Round 1

Reviewer 1 Report

Dear authors,

please find my comments in attachment.

Sincerely,

the reviewer

Reviewer 2 Report

This paper brings new knowledge on the flora and vegetation characteristics , phytogeography and conservation value of the montane coniferous forests of Central Balkan Peninsula. The authors did a very extensive research, comprising a broad geographical range of these forests. The methodology is well described, with lot of data used in the analyses. The authors pointed out conservation values of these forests in the context of the EU Habitats Directive, for the first time they calculated and compare level of endemism, and identify the centers of floristic richness of the coniferous forests in Europe. All results significantly contributes to better understanding of these habitats which is an important step in planning the conservation measures.

In the Word document there are suggestions and comments given by the Reviewer.

Round 2

Reviewer 1 Report

Dear authors,

the article has been improved, I appreciated your efforts. The justifications proposed for the flaws highlighted are sufficient, therefore I have no further observations.

Sincerely,

the reviewer